# A Review of the Effectiveness, Feasibility, and Acceptability of Art Therapy for Children and Adolescents during the COVID-19 Pandemic

**DOI:** 10.3390/ijerph191811612

**Published:** 2022-09-15

**Authors:** Minh Ngoc Le Vu, Anh Linh Do, Laurent Boyer, Quy Chi Tran, Stefan Kohler, Syed Ishtiaque Ahmed, Andreea Molnar, Tung Son Vu, Nhan Trong Huynh Vo, Linh Mai Vu Nguyen, Linh Gia Vu, Vu Anh Trong Dam, Thomy Duong, Dan Linh Nguyen Do, Ngoc Minh Do, Roger S. Mclntyre, Carl Latkin, Roger Chun Man Ho, Cyrus Su Hui Ho

**Affiliations:** 1Institute for Global Health Innovations, Duy Tan University, Da Nang 550000, Vietnam; 2Institute of Health Economics and Technology, Hanoi 100000, Vietnam; 3Research Centre on Health Services and Quality of Life, Aix Marseille University, 13385 Marseille, France; 4High School for Gifted Students, Hanoi University of Science, Hanoi 100000, Vietnam; 5Heidelberg Institute of Global Health, Heidelberg University, 69120 Heidelberg, Germany; 6Department of Computer Science, University of Toronto, Toronto, ON M5T 2S8, Canada; 7Department of Computer Science and Software Engineering, School of Software and Electrical Engineering, Swinburne University of Technology, Melbourne, VIC 3122, Australia; 8Faculty of Medicine, Duy Tan University, Da Nang 550000, Vietnam; 9Vinschool Education System, Hanoi 100000, Vietnam; 10Institute of Medical Sciences, University of Toronto, Toronto, ON M5T 2S8, Canada; 11Bloomberg School of Public Health, Johns Hopkins University, Baltimore, MD 21205, USA; 12Department of Psychological Medicine, Yong Loo Lin School of Medicine, National University of Singapore, Singapore 119228, Singapore; 13Institute for Health Innovation and Technology (iHealthtech), National University of Singapore, Singapore 119077, Singapore

**Keywords:** feasibility, acceptability, art therapy, children, COVID-19, epidemic, mental health

## Abstract

Art therapy has been widely offered to reduce symptoms of psychological disturbance. Pooled evidence about its effectiveness in epidemic contexts, particularly during the COVID-19 pandemic, has not been yet established. This study reviewed the effectiveness, feasibility, and acceptability of art therapy on children and adolescents during the COVID-19 pandemic and past epidemics. We searched PubMed/Medline, PsycINFO, CENTRAL (Cochrane Library), and CINAHL for articles on art therapy during COVID-19. Included studies reported improvements in measures of mental health, sleep quality, and psychological well-being in children with or without disabilities in the epidemic context. Results also showed that art therapy was highly feasible and accepted by children and adolescents as well as their families during epidemics in reviewed studies. Art therapy can be effective at improving various aspects of mental health, sleep quality, and psychological well-being. More empirical evidence is needed with larger sample sizes and longer duration of interventions.

## 1. Introduction

The COVID-19 pandemic is recognized as a global public health crisis as well as a mental health crisis [1]. Children and adolescents are especially vulnerable to social disruptions caused by COVID-19 control measures [2]. Not only did the COVID-19 pandemic exert negative impacts on the well-being and mental health of children directly but also indirectly via the implementation of public health interventions for COVID-19 such as school closure and lockdown. Along with the socio-economic burden, the lack of social interactions and appropriate support systems during COVID-19 has further amplified mental health risks, especially among the younger population [3].

Extant literature has demonstrated that children and adolescents experienced mental health problems related to the COVID-19 pandemic. A number of mental conditions were documented during the time of the pandemic: increased measures of fear, anxiety, depression, suicidality, sleep disturbance, post-traumatic stress disorder (PTSD)-related symptoms, as well as drug and alcohol misuse coinciding with school closures [4,5,6]. Other reported psychological conditions caused by COVID-19 included clinging, inattention and irritability, decreased appetite, fatigue, and discomfort [7]. Furthermore, increased stress related to family life had also been reported among both children and parents and this further dampened the parent-children worsening relationship during the COVID-19 pandemic; examples of stressors include parents’ pressure from home childcare during social lockdowns, family members’ admission to quarantine and/or hospitalization or even deaths resulted from COVID-19 infection, children’s lack of access to resources of mental health care such as telepsychiatry [3,8]. Increased risk of exposure to domestic violence, abuse, and neglect during COVID-19 was also noted; lack of contact with people, who would normally notice the mental consequences of such maltreatment, posed further challenges in improving the mental state of children and adolescents [8]. Besides immediate effects on mental health, the aftermath of COVID-19 poses tremendous impacts on the long-term development of children and adolescents. For instance, Liu et al. discussed that stable companionship is among the core of a child’s healthy psychological development, and thus separation from parents, both short-term and long-term, due to COVID-19 presents adverse changes in their mental health [9]. Salient issues encountered by parental loss and isolation from parents in previous settings include heightened risk of mood disorders and psychosis as well as latent depression and anxiety in adulthood [10].

Art therapy is defined as a form of psychotherapy that includes the use of art as its main mode of expression and communication [11]. A wealth of studies have strived to identify the root of art therapy and its development trajectory. By origin, art therapy was utilized for treating impairment in symbolic systems or trauma-induced troubles in behavioral development [12,13]. It is distinguished from other forms of therapy in the unique benefits of soothing mental issues through methodological use of lightweight yet influential art means such as music, drawing, painting or sculpting with an aim to induce changes, progress and acceptance in the mental state of the receiver [14]. Therefore, although the event of a pandemic or disaster rarely results in such extreme disability, art therapy is still considered a potential intervention to improve the mental well-being of children and adolescents [15,16]. During COVID-19, most common art therapies include music-, visual art-, movement-, and expressive-writing-based therapies [17].This approach allows children to use art as a form of emotional expression, and at the same time enhances psychosocial care for children during medical treatment and recovery [18]. Art therapy as mental health treatment has been widely used and successfully applied in different healthcare and non-healthcare settings [19]. An example of art therapy used to improve mental health is art therapy telehealth, which has been reported to promote emotional stabilization, a sense of connection, and self-worthiness and foster engagement, integration, and participation of family and community members [20,21].

Previous reviews of the effectiveness of art therapy in different populations of children suggest that art therapy is effective in improving their physical and psychological health as well as their quality of life [22,23,24]. During COVID-19, the use of art therapy underwent major changes in its implementation, specifically through online platforms and on a larger scale instead of private in-person therapy sessions. This characteristic is a prime advantage during the context of a public health crisis, as services can be provided online, and thus can be available wide-scale to reach a larger number of children. Therefore, the objective of this study is to assess the effectiveness, feasibility and acceptability of art therapy among children during COVID-19 as well as identify knowledge gaps to be fulfilled for future settings. This rapid review aims to inform therapists and care providers with a comprehensive picture of art therapy globally to develop best-fit art therapy services and art-based rehabilitation programs for their community, especially after COVID-19.

## 2. Example Implementations of Art Therapy

During COVID-19, art therapy was delivered in several forms, most common of which were drawing, music listening and playing, narrative interactions [25]. Although all were generally effective, their impacts ranged from mild mood enhancement to major improvement of sleep quality or reduction of hyperactivity. In this review, we investigated three main art therapy interventions to assess their effectiveness, feasibility and acceptability among children and parents. Table 1 summarized selected interventions of art therapy during COVID-19 including their settings, delivery, measures and outcomes.

## 3. Effectiveness of Art Therapy

Malboeuf-Hurtubise et al. conducted a randomized cluster trial in Quebec, Canada during the COVID-19 lockdown period [28]. The authors employed two groups comprising 14 school children who received emotion-based directed drawing intervention in the first group and 8 children who received mandala drawing intervention in the second group. Both groups underwent five weekly 45-min sessions during which children were asked to make drawings and discuss their works. Contents of the drawings in the first group were child participants’ feelings and COVID-19-related mental issues (e.g., fear, worry, irritation, etc.); meanwhile, the second group drew mandala only for each session, with no COVID-19-related exercise being given. While the authors could not identify any post-intervention differences regarding inattention, anxiety, depression, hyperactivity, and mindfulness between groups 1 and 2, they observed that across the entire sample, after the post-hoc sensitivity analysis, hyperactivity scores among children showed a significant reduction, suggesting that drawing-based interventions could provide mental health improvement [28].

Bombard et al. and Cho et al. explored how using music therapy could stimulate positive feelings among children during the COVID-19 pandemic [26]. One study provided specifically curated music playlists to 19 mothers and instructed them to play the music to children during the week, especially when their children were in a negative mood and in need of improvement, or when in a positive mood but in need of prolonging such a state. The study reported that 81.2% of mothers reported that children reported themselves as happier and experienced less psychological distress after listening to the music, while 13.7% of mothers did not find any effects of the intervention on children’s mood [26]. Another study [27] used home-based music therapy aiming to improve sleep quality, and created personalized visual soundtracks for children with developmental disorders, whose rehabilitation therapies were disrupted by COVID-19. Among 12 children who participated, the authors observed that sleep breathing disorders, sleep-wake transition disorders, and overall sleep disturbance improved after the intervention (*p* < 0.05) [27]. 

## 4. Feasibility and Acceptability of Art Therapy

Regarding the feasibility, of offline interventions, the material preparation seemed to be the most complicated stage. For example, Sarah Bompard et al. (2021) emphasized that for children with developmental disorders, visual soundtrack preparation required the involvement of a music specialist who completed clinical training in music therapy and held a degree in Education; persons with such combined backgrounds might be hard to find for similar studies [27] (Table 2). For online interventions, Malboeuf-Hurtubise et al. (2021) noted that the intervention was highly feasible with few technical complications, such as a secured video call platform. Requirements for instructors who had experience in mindfulness-based intervention and art-therapy knowledge to organize drawing were mentioned [28]. Cho & Ilari (2021) found that they were unable to control participants’ engagement through virtual intervention; and they suggested that increased social presence during the study (e.g., via visual representation of researchers) might help improve and sustain the motivation of participants [26].

In terms of acceptability, all trials found that children and facilitators engaged in the intervention and, to a certain extent, accepted to continue performing intervention-related activities even after the intervention ended, suggesting that there is a high acceptance level of art therapy for children in the epidemic context [16,26,27,28].

## 5. Changes in COVID-19 Art Therapy Delivery

A key difference in delivering art therapy interventions during the COVID-19 pandemic when compared to the normal context or the previous Ebola epidemic is the use of virtual video conference tools due to local social distancing requirements since two out of three interventions were implemented via this platform. The core advantages of this approach in the context of epidemic are that art therapy is easily accessible and there are several types of art-based interventions available that are inexpensive and capable of large-scale implementation with high level of acceptability by children and their families. When children and adolescents stay at home due to infectious epidemics can participate in virtual art therapy sessions via the Internet, art therapists can provide their therapy services to a greater percentage of the healthcare-seeking population and address an important gap in the healthcare system [28]. Videoconferencing technology enables art therapists to have timely access to a larger number of children and adolescents who experience disruptions in mental healthcare and allow children to interact in a group during highly collective activities. The use of technology to deliver online art therapy sessions has also been shared in two case studies in the United Kingdom and the United States, to healthy children and those with anorexia nervosa during the COVID-19 pandemic [29,30]. 

As the stress induced by COVID-19 shares many similarities with previous pandemics, implications can also be drawn from discussed settings. However, practitioners need to make suitable adaptations due to unique characteristics of COVID-19. Firstly, to successfully implement an art therapy intervention during the COVID-19 pandemic and future epidemics, the content should consider children’s culture and abilities. Secondly, further virtual art therapy interventions should be warranted and tested among children in quarantine areas who were particularly vulnerable to mental disorders [6], or children whose parents were frontline health workers since none of the interventions were performed for this population. Several knowledge gaps in art-based interventions need to be addressed, for instance, evidence from wide-ranging art therapies or responses from diverse-background populations to art-based interventions.

## 6. Quality Assessment and Ethical Considerations

The methodological quality of the studies was assessed by using the Mixed Methods Appraisal Tool, which examines the methodological quality of qualitative research, randomized controlled trials, nonrandomized studies, quantitative descriptive studies, and mixed-methods studies [31]. As this review used published and secondary data, no ethical review was thought of.

## 7. Limitations

This review has several limitations. Firstly, although we included major types of art therapy in the inclusion criteria, we only found four studies for analysis. Therefore, it is possible that some relevant articles might not be retrieved during our search process despite using search strategies and iterative stages to minimize omissions. Secondly, we restricted our search to the English language and scientific articles; thus, studies published in other languages or other document forms (e.g., grey literature, reports, etc.) could be missed. Moreover, due to the relatively nascent area, there is insufficient data to characterize aspects of the setting, training requirements, outcome measures, and how best to personalize art therapy treatment approaches. 

## 8. Conclusions

Art therapy is feasible a complementary therapy and can help individuals to overcome the psychological conditions during COVID-19 should it be integrated into schools, community-based activities and family supports.

## Figures and Tables

**Table 1 ijerph-19-11612-t001:** Summary of the selected papers.

Authors (Year)	Settings, Time	Epidemic	Type of Art Therapy	Study Descriptions	Measures and Outcomes
Catherine Malboeuf-Hurtubise et al. (2021)	Quebec, Canada, May–June 2020 when performed province-wide lockdown.	COVID-19 pandemic	Drawing	A randomized cluster pilot trial was performed on 22 elementary school children (4th and 5th grades), mean age = 11.3 years. Group 1 received emotion-based directed drawing intervention in 5 weeks, 45 min/ session/week to explore the emotion (e.g., fear, worry, irritation, etc.) and/or discuss COVID-19-related issues (*n* = 14). Group 2 received mandala drawing intervention. Students only drew mandalas without instructions and COVID-19-related activities, following discussions about drawings (*n* = 8). Baseline and follow-up surveys were performed one week before and after interventions, respectively.	Behavior Assessment Scale for Children-3rd edition: No differences were found in anxiety, depression, and hyperactivity between groups 1 and 2. Inattention was improved in group 1 compared to group 2; however, after adjustment, the difference was not significant. Levels of hyperactivity in entire samples reduced significantly after interventions (p < 0.05).
					Mindful Attention Awareness Scale for Children: No difference was found in mindfulness between groups 1 and 2.
Eun Cho et al. (2021) [26]	United States, August 2020	COVID-19 pandemic	Music listening	A trial was performed on 19 mothers (age range 30–39 years). They have received music playlists and considered using specific playlists for each of two moments: (1) when their children were in bad mood, and (2) when their children were in a positive mood to improve their children’s well-being.	81.2% reported that children seemed to be happier and had less psychological distress after listening to the music, while 13.7% did not find any effects of the intervention on children’s mood. Mother-children interaction improved at home.
Sarah Bompard et al. (2021) [27]	Italy, 2020	COVID-19 pandemic	Euterpe method’s music therapy	A trial was performed on 12 children with developmental disorders (age below 12 years old). They received personalized home-based music therapy entitled the “Euterpe” method combined between a visual soundtrack and multisensory stimulation. The audio file included four parts: (1) introduction (40 s), (2) maternal voice + patient’s vocal reaction; (3) most effective track, and (4) lullaby song by the mother. Children watched the file three times a day for 12 consecutive days.	Sleep Disturbance Scale for Children: Sleep breathing disorders, sleep-wake transition disorders and overall sleep disturbance among children improved after treatment (*p* < 0.05).
					Parenting Stress Index-Short Form: Parental distress and Defensive responses among parents improved after treatment (*p* < 0.05)

**Table 2 ijerph-19-11612-t002:** Qualitative synthesis for feasibility, acceptability, and limitations in selected studies.

Authors (Year)	Epidemic	Type, Setting	Feasibility	Acceptability	Limitations in Implementation	Recommendation for Implementation
Catherine Malboeuf-Hurtubise et al. (2021) [28]	COVID-19	Drawing, Online	Materials: Few technical barriers. Secured video conference platform with password. Materials such as guidelines about drawing on specific topics like COVID-19.	Teachers were motivated to involve in the intervention. Students well accepted and appreciated the intervention.	Not available	Not available
			Facilitators’ required experience: Psychology instructors with experience in mindfulness-based intervention and communication with children. Structured clinical supervision provided by pediatricians.			
			Children’s ability: Grade 4th or 5th. Children know each other well. Attend online drawing classes. Involve in group discussion.			
[26]	COVID-19 pandemic	Music listening, Online	Materials: Few technical barriers. Should carefully prepare music playlists and instructions.	More than half of the children actively reacted to the music (singing, moving body, dancing). All mothers showed a positive attitude toward interventions.	Unable to control participants’ engagement in the intervention	Careful and deliberate plans should be considered to increase the social presence to improve and sustain the motivation of participants. Video presentations of the researchers might be helpful.
			Facilitators’ required experience (mother): No previous experience required. Had basic Internet use skill. Read and follow instructions.			
			Children’s ability: No requirement.			
[27]	COVID-19 pandemic	Euterpe method’s music therapy, Offline	Materials: Visual soundtrack preparation required the involvement of music therapists and experienced composers.	All families agreed to continue participating in the therapy.	Not available	The development of the soundtrack should have the involvement of the mother.
			Facilitators’ required experience: Parents should have basic experience in using these files.			
			Children’s ability: No requirement.			

## Data Availability

Not applicable.

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
