# Peer review of "A Review of the Effectiveness, Feasibility, and Acceptability of Art Therapy for Children and Adolescents during the COVID-19 Pandemic"

_ijerph, 2022, doi:10.3390/ijerph191811612_

Round 1

Reviewer 1 Report

The manuscript provided by Le Vu et al. showed that art therapy can be effective at improving various aspects of mental health, sleep quality, and psychological well-being. The results of this paper provide an insight into art therapy for psychological well-being. I have provided my comments to author as follows:

·      Authors used limited number of search engines to find out the manuscript related to art therapy and psychological conditions induced during COVID-19 periods. These limited number of resources resulted in smaller size of manuscripts found by authors. So, I recommend authors to expand the search strategy to other sources and language.

·      Authors did not provide the exact search terms used for finding and extracting the manuscript.

·      Figure 1: Flow chart diagram has lots of issues. 1. Authors did not mention the no. of studies excluded at the level of duplication check. 2. A phrase catch my notice beyond the lower right box, which cannot be seen clearly. Overall, authors should list all manuscripts after duplication check and provide the reasons why did they decided not to review those papers, in a supplementary table.

·      Authors should check the risk of bias of each manuscript involved in the study and clearly mentioned the reason why they chose as high-risk, low-risk, or unclear judgment.

·      The aim of the current study is to evaluate the benefits of art therapy in psychological conditions induced during COVID-19. It is not acceptable to involve a study related to other disease and condition. For example, a study conducted by C. A Decosimo evaluate the outcome of a community-based psychosocial expressive arts program for children during the Liberian Ebola epidemic.

·      The overall conclusion of this study is as follows “Art therapy is a good therapy and can help individuals to overcome the psychological conditions during COVID-19”. This is not a professional/academic way to conclude. Everyone knows art therapy is an effective therapy on mental disorders. Authors should re-write the conclusion in any other way. For example, “Our systematic review showed that the most effective way is …” or “… the most feasible or effective way is …”.

·      Several minor linguistic errors and typos have been observed, please re-checked the manuscript to resolve all the errors.

Author Response

Dear reviewer,

Thank you very much for your valuable suggestions. We have made adjustments to our manuscript as follows:     

1. Authors did not provide the exact search terms used for finding and extracting the manuscript.

Response: We have decided to change this manuscript to a literature review instead of a systematic review.

2. Figure 1: Flow chart diagram has lots of issues. 1. Authors did not mention the no. of studies excluded at the level of duplication check. 2. A phrase catch my notice beyond the lower right box, which cannot be seen clearly. Overall, authors should list all manuscripts after duplication check and provide the reasons why did they decided not to review those papers, in a supplementary table.

Response: We have decided to change this manuscript to a literature review instead of a systematic review.

3. Authors should check the risk of bias of each manuscript involved in the study and clearly mentioned the reason why they chose as high-risk, low-risk, or unclear judgment.

Response: We have decided to change this manuscript to a literature review instead of a systematic review.

4. The aim of the current study is to evaluate the benefits of art therapy in psychological conditions induced during COVID-19. It is not acceptable to involve a study related to other disease and condition. For example, a study conducted by C. A Decosimo evaluate the outcome of a community-based psychosocial expressive arts program for children during the Liberian Ebola epidemic.

Response: We have excluded the study conducted during Ebola and discussed the characteristics of art therapy in COVID-19.

5. The overall conclusion of this study is as follows “Art therapy is a good therapy and can help individuals to overcome the psychological conditions during COVID-19”. This is not a professional/academic way to conclude. Everyone knows art therapy is an effective therapy on mental disorders. Authors should re-write the conclusion in any other way. For example, “Our systematic review showed that the most effective way is …” or “… the most feasible or effective way is …”.

Response: We have re-wrote our conclusion and made adjustments to our discussion.

6. Several minor linguistic errors and typos have been observed, please re-checked the manuscript to resolve all the errors.

Response: We have reviewed our manuscript to fix linguistic errors. 

On behalf of our team, thank you for your consideration and we look forward to hearing from you.

Reviewer 2 Report

Dear Authors,
thank you very much for the opportunity to read this article. I think, that the topic of this article is very interesting. Unfortunately, I do not recommend the article for publication in this form. I have three major issues:
1. Art therapy definition. The theoretical explanation of Art Therapy is insufficient. In the scientifical literature, there are two approaches to the definition of Art Therapy (it is advisable to list the types of Art Therapy).
2. Epidemie. I also see the part of the title "previous epidemics" (and thus the inclusion of the article Decosimo et al. (2019)) as questionable. This (previous epidemics) is not sufficiently justified both in the theoretical introduction and methodologically too.
3. Systematic review. This is not a systematic review, but a literary review. If it were to be a systematic review, a meta-analysis should be done.
Recommendation:
- improve the theoretical base of the introduction - clearly state the characteristics of art therapy which is used as a theoretical background
- define an "epidemic" - between COVID and Ebola (in terms of global impact, there is a significant difference)
- either rename it to a literature review or add a meta-analysis.

Author Response

Dear reviewer, 

Our team would like to thank you very much for the valuable suggestions you proposed. We have made major adjustments to our paper in light of your comments:

1. Improve the theoretical base of the introduction - clearly state the characteristics of art therapy which is used as a theoretical background

Response: We have added definition, theoretical base and characteristics of art therapy in the introduction.
Location: Line 90 - 108 p.2

2. Define an "epidemic" - between COVID and Ebola (in terms of global impact, there is a significant difference)

Response: Our team has decided to narrow the study to the COVID-19 pandemic and no longer included the study during Ebola.

3. Either rename it to a literature review or add a meta-analysis.

Response: We have changed the paper to a review.

On behalf of our team, thank you for your consideration and we look forward to hearing from you.

Round 2

Reviewer 1 Report

Authors have change the design of the study, so my comments have been will be all addressed automatically. Although the value of the manuscript has now been decreased, as review manuscript, this is well-done/design paper. I do not have any additional concerns on this paper.

Reviewer 2 Report

The authors revised the article according to the comments

The article is publishable in this form.